# Health Status and Access to Healthcare for Uninsured Migrants in Germany: A Qualitative Study on the Involvement of Public Authorities in Nine Cities

**DOI:** 10.3390/ijerph19116613

**Published:** 2022-05-28

**Authors:** Lukas Kratzsch, Kayvan Bozorgmehr, Joachim Szecsenyi, Stefan Nöst

**Affiliations:** 1Department of General Practice and Health Services Research, University Hospital Heidelberg, 69120 Heidelberg, Germany; lukas.kratzsch@uni-heidelberg.de (L.K.); kayvan.bozorgmehr@uni-bielefeld.de (K.B.); joachim.szecsenyi@med.uni-heidelberg.de (J.S.); 2Department of Population Medicine and Health Services Research, School of Public Health, Bielefeld University, 33501 Bielefeld, Germany; 3Faculty of Business and Health, School of Health Sciences and Management, Baden-Wuerttemberg Cooperative State University Stuttgart, 70178 Stuttgart, Germany

**Keywords:** health services accessibility, health inequality, public authorities, transients and migrants, uninsured migrants, irregular migrants, EU citizens, Germany, qualitative research, health services research

## Abstract

Non-governmental organisations (NGOs) regularly report data on their work with uninsured migrants (UM) within a (so-called) parallel health care system. The role and involvement of public authorities therein have yet been underrepresented in research. Our aim was to gain a better understanding of public authorities’ role in the parallel health care system and their view of the health situation of UM. We conducted qualitative semi-structured interviews with 12 experts recruited by purposive sampling from local public health authorities (LPHAs), state-level public health authorities (SPHAs), and social services offices (SSO) in nine cities, recorded, transcribed, and subjected the data to qualitative content analysis. LPHAs are more often directly involved in providing medical services, while SSOs and SPHAs function as gatekeepers for access to social benefits, including health insurance, and in grant-funded projects. NGOs keep substituting for the lack of access to regular health care from public institutions, but even in settings with extended services, public authorities and NGOs have not been able to provide sufficient care through the parallel health care system: Experts report gaps in the provision of health care with respect to the depth and height of coverage, due to the fragmentation of services and (ostensible) resource scarcity. Our study highlights the necessity for universal access to regular health care to overcome the fragmentation of services and improve access to needed health care for UM in Germany.

## 1. Introduction

Most people in Germany are covered by a social health insurance system, which is financed through loan-based contributions, while social welfare services cover the contributions of unemployed persons. Germany spends more on health than most other European countries and allegedly provides universal coverage [1,2]. Official statistics confirm good overall coverage, estimating only around 61,000 people *without* health care coverage in 2019 [3,4]. This estimation is flawed since populations such as irregular migrants and homeless people are not reached through the methodology of a micro-census, which is a national household survey covering about 1% of the national population [5]. Simultaneously, especially these migrant populations are potentially at risk of experiencing important access barriers to health insurance and health care services [2]. From the perspective of health care providers in Germany, this heterogeneous population is perceived as uninsured migrants (UM), comprising irregular migrants (e.g., “sans papiers” or undocumented migrants) from non-European countries, as well as EU migrants without claims to social benefits or lack of previous coverage [2,6,7,8]. UM must gain access to health care services through parallel structures set up outside the realm of the regular health care system [4,9,10]. In these resource-scarce settings, NGOs and local public health authorities (“Gesundheitsamt”, LPHAs) professionals and volunteers are involved in the organisation and the provision of care. To some extent, social security offices (“Sozialamt”, SSOs) are also involved, as they present the access point to social welfare, including granting health insurance, as well as providing health care vouchers and funding i-patient treatment. Existing data on UM, including health status and access to care, are not representative. They are only available in the form of heterogeneous reports of service-utilisation from NGOs or public authorities, with strong variation in population characteristics (i.e., EU migrant vs. non-EU migrant) due to variation in local settings [11,12,13,14]. Quantitative and qualitative studies have shown that the role of LPHAs in these settings is very diverse and that important gaps in the provision of health care remain [15,16,17,18,19]. For example, even in the framework of binding national legislation mandating the treatment of communicable diseases according to the Infection Protection Act (“Infektionsschutzgesetz”, IfSG) [20], gaps in de facto access to needed health services remain [16]. However, evidence suggests that when needs are identified and communicated, the care situation on the ground can improve with the help of public services [21], as public funding represents the necessary transition from voluntary work and donation-financed projects to publicly financed services. 

In the scope of the MONITORaccess study [22], the NGO perspective on the health monitoring of UM was evaluated through focus groups, aiming to develop a tool for health status and health care monitoring of this population. Following up on this, we explored the perspective of public (health) authorities regarding (i) the health needs and health status of UM; and (ii) the local health care situation of UM. We further sought to understand the role and involvement of public (health) authorities in the provision and planning of health care for UM, with a particular focus on the type of services provided and their role in community-level networks and institutional arrangements.

## 2. Materials and Methods

### 2.1. Study Design

A qualitative approach with semi-structured expert interviews was chosen to explore the perspectives of experts from LPHAs, state-level public health authorities (SPHAs), and SSOs on the parallel health care system. From prior research and work in the field, it is known to the authors that certain public authorities seem to be more involved than others and motivation to participate in research was especially high in more involved authorities. Additionally, some public authorities have refrained from participating in studies on this topic previously [5]. The sampling method chosen was, therefore, purposive sampling, aiming to recruit experts from LPHAs with services for UM, other than anonymous testing for infectious diseases and SSO services that we knew were involved in the care for UM. Experts were defined according to expected knowledge gain [23,24] as employees of the authorities that are directly or indirectly (e.g., coordination of services) involved in the provision of health care to UM. We aimed to include experts from cities of different sizes, knowing that the target population is likely more prevalent and visible in urban areas [15,25]. We included experts from across Germany to explore a variety of different care settings.

### 2.2. Sampling and Data Collection Methods

Purposive sampling was carried out with the aim of including a broad spectrum of different types of relevant public authorities. This process was therefore based on selected criteria of the legal grounds of the Infection Protection Act (“Infektionsschutzgesetz”, IfSG) [20], the state (“Länder”), laws on public health services [26], the access options to health services from the social security codes [2], as well as pre-existing knowledge on the connection of authorities to care settings. In total, 18 public authorities were contacted first by telephone and then by sending a standardised request via email. Further, two experts were added through snowball sampling. After contacting all the authorities, eleven interviews were finally agreed upon. Of these, three took place in social welfare offices and eight in public health offices. One interview was conducted with two experts at the same time, sharing one work position. Seven public authorities declined interviews, mostly due to time constraints, and one stated to decline research requests in general. The interviews took place between July 2017 and January 2018. Regarding informed consent, an audio recording was made of each interview, and a post-interview protocol was created. Personal and professional characteristics on age, education, official position title and previous time in the authority were collected by questionnaire. In view of the state of research and taking into account the authors’ research experiences and the epistemological interest of this study, topics for an interview guide were collected, prioritised, and sorted through several discussions within the research group. The topics of the final interview guide were visualised as a mind map (Figure 1 and Appendix A) that enables openness as well as an in-depth thematic study while maintaining comparability (Table 1). Relating to the main topics of the interview guide, the interviews were typically discussed in this order: An introductory question about the daily work routine and its relation to UM, the local care situation and local networks, challenges of care on-site, health status, and access barriers of the target population, as well as data work in the authority.

### 2.3. Data Analysis

All of the data were pseudonymised in the interview transcript so that all personally identifying data were obliterated, and no inferences could be made about individual persons or organisations within the text data, except for LK. Following the interviews, the audio recordings were transcribed into MAXQDA 12 in a rule-guided manner. A post-interview transcript was prepared for each interview [27]. Because of a technical issue in interview 1 (P1), only parts of it were clearly registered. With the help of the handwritten notes, a summary reconstruction of the statements was created and used as a basis for the analysis. A dual evaluation method, concluding inductive and deductive steps [24,28], was designed to examine the data in terms of information generation as well as an interpretation of the relevant systems and motivations. The data were inductively divided into categories (topics) and deductively coded in parallel. Grouped and subsumed, main and sub-categories were created in the research process to structure and extract the text passages from the original material. The excerpts (about 600 text segments in the German language) were sorted in table form and then each summarised. In the next step, we analysed each local care setting in line with the codes on the case level and then on the inter-case level. (See Appendix A for a translated example of this process). As a final step, we contextualised the findings according to the existing research and known grey literature where available. The material was evaluated according to the main categories “personal view of the care situation of the target group” and “significance in the everyday work of the authorities”. The care setting was analysed according to three dimensions: *breadth* (the extent of the population covered), *depth* (the number and type of services covered), and *height* (the extent to which the costs of services are covered by prepaid financing) [29,30]. This study mainly focuses on the depth and height of coverage.

LK, who conducted the interviews, is a resident doctor of internal medicine and used to be active as a volunteer in an NGO organising access to healthcare for uninsured migrants. In order to take the researcher’s perspective into account, a series of assumptions were formulated prior to conducting the interviews, which were reflected upon within the research group after the first round of coding. LK, SN, and KB reflected on the focus of the preliminary findings regarding bias toward an activist’s specific perspective. As a result, we adjusted the coding and coded the data repeatedly.

Ten out of the 12 experts interviewed work on the communal level, while two on the state level of government. For the inter-case comparison of local settings, we included settings 1–7 for evaluation by excluding state-level experts and settings where we only interviewed one expert from social services offices (Settings 8 and 9). All of the settings were included for the rest of the analysis. We categorised the settings by the information given in the interview and added further information from our à-priori research on the local settings where needed, and created graphical overviews for each case to visualise local networks and interactions (Figure 2 and Appendix A). We did not intend to do a systematic network analysis but wanted to create a visual aid in addition to the code-driven analysis. The rationale for the described procedure resulted from the dual research claim of systematising the experts’ opinion against the backdrop of the main evaluation categories and the exploration of local dynamics of care setting development. The reporting of the results was conducted in accordance with the COREQ reporting guidelines [31].

## 3. Results

### 3.1. Interviews and Participants

The expert sample consists of nine women and three men. Four of them are physicians, two are from the field of social work/social education, five are from the field of public management/administration, and one expert is from the field of public health. The experts work in authorities from eight different German federal states and nine different cities. Two of the experts work in public health authorities that are regulated at the state level (SPHA). One expert works in an SSO of a city-state and is thus counted as communal for the analysis. Of the communal authorities, most experts (n = 7) are from a metropolis (>500,000 inhabitants), and three are from large cities (>100,000 inhabitants). Ten out of eleven had worked in a public authority for more than two years (mean = 12.3 years, median = 8.3 years). The interviews were conducted on the premises of the respective authority and lasted between 45 min and two hours.

### 3.2. Role and Involvement of Local Health Authorities in Individual Health Services

Concerning services for UM, of the seven LPHAs in the sample, all provide health or medical counselling, diagnostics and, at least on a case-by-case basis, treatment of infectious diseases within the framework of the Infection Protection Act (Table 2). They also offer pregnancy counselling (n = 5), including pregnancy conflict counselling (“Schwangerschaftskonfliktberatung”), gynaecological examinations, including antenatal care (n = 4), and co-finance and organise deliveries (n = 2). One expert is regularly exempted from his/her duties to provide medical consultation hours at a local NGO. Two agencies have socio-legal clearing on-site, although one is predominantly for women. One authority funds socio-legal clearing by an NGO. One agency funds the maintenance and treatment costs of a local NGO drop-in centre, and two LPHAs have drop-in medical centres on-site. The experts from an SPHA were solely involved in coordinating projects for UM and did not work directly with the target population in personal services. The LPHAs studied can be roughly divided into two groups: those that offer multiple of the before mentioned services (n = 4) and those that primarily offer services under the IfSG (n = 3).

### 3.3. Role and Involvement of Social Serive

Experts from SSO are indirectly linked to the health care of UM through their involvement in steering committees where they collaborate with NGOs and the LPHA in the administration and modification of grant-funded projects for UM. In the role of experts in working groups concerning UM, they share and apply their expertise on the legal framework to enable UM to gain insurance coverage. Otherwise, they process applications for reimbursement of expenses by hospitals (i.e., §25 Social Security Code XII) when uninsured patients have, e.g., received urgent in-patient treatment. SSOs can theoretically grant vouchers to UM for access to the regular health care providers, a service that is practically impossible to use for illegalized migrants, as legal status is transmitted to immigration authorities according to §87 of the Residence Act (“Aufenthaltsgesetz” AufenthG) [32]. This can lead to legal repression and possible expulsion. The conflict between the theoretical entitlement to this service and the legal requirement of reporting is illustrated by an expert from an SSO:

“Of course, we can’t say that much about illegal immigrants, who are usually people who have applied for asylum at some point and have been rejected, who must actually leave the country or be expelled, and who then go into hiding. Of course, they don’t dare to come to us as authority because they know that the moment they arrive here, we would have to inform the immigration authority that someone is here illegally.”(P5)

One expert explained that UM’s health-related issues constitute a very small part of the work of social services and are described as “marginal” in their daily routine, while another named EU citizens as their main target population, although access to health care was secondary to general access to social benefits. The expert, P5, describes her perspective on the discrepancy between legal theory and her experience with clients:

“The theory is that, if you believe the politicians, everyone is theoretically insured, that’s how it looks at first, but in practice, unfortunately, it looks quite different because the assumption that people from abroad bring [a health] insurance with them and therefore have this EHIC [European Health Insurance Card] is often not true. So for us this is a big problem, first of all that we are confronted again and again with the people and of course we know exactly if we do not grant any benefits there is usually no insurance coverage.”(P5)

As their work does often not comprise direct contact with UM but is rather an administrative context, the challenges reported are thus administrative and, to a lesser extent, a humanitarian matter. The main challenges described are the complex legal framework for EU citizens to gain access to social welfare benefits and to reimburse hospitals due to a lack of information on the case in question. The threshold of access is further increased by complicated legal fragmentation, prohibiting or allowing access to social welfare. Expert P5 concludes: 

“So there are all kinds of parallel structures that should actually lead to everyone being insured, which are first of all totally complicated, and besides, it’s not like everyone is insured, so there are still enough uninsured. And I think that the German state can’t afford to make such a mess <laughs>.”(P5)

### 3.4. Reported Themes of Challenges

#### 3.4.1. Services in the Framework of the Infection Protection Act

Counselling and diagnosis of diseases within the framework of the IfSG are described as quasi-standard offers of disease management in all LPHAs. However, the treatment of infectious diseases is described as difficult in some cases, even in LPHAs that provide extended services, using HIV/AIDS and hepatitis C as examples (Table 3). The main challenge is the financing of therapies due to complex case constellations for UM: If no access to insurance coverage can be established, solutions mentioned include temporary the suspension of expulsion (“Duldung”) due to humanitarian reasons for the time applicable (e.g., time of treatment) and temporary migration to an EU country with easier access to health care or return to the home country (for EU citizens). In the absence of such makeshift solutions, the therapy costs are partly covered in individual cases by the LPHA, i.e., the treatment of tuberculosis. Experts describe additional makeshift solutions that are attempted in cooperation with NGOs or doctors in private practices, based on goodwill:

“Who pays for the medication costs, which can reach hundreds of euros? My colleague also has certain collaborations with doctors where it is always possible to care for someone, but these are really very small, isolated solutions. If we had more, I don’t know how it would be. […] You cannot pay for HIV medication as an NGO or as a city for the entire life.”(P2)

#### 3.4.2. Pregnancy

Antenatal ambulant care is provided in two LPHAs by gynaecologists in collaboration with social workers who try to achieve the integration of the client into social welfare and regular health care if possible. One of the structured maternity programmes for pregnant women lets patients make down payments during pregnancy so that delivery can take place at a reduced price of EUR 600 on the date of birth in cooperation with local hospitals. If a patient cannot afford the sum, local NGOs or, exceptionally, the LPHA will cover the costs. Elsewhere, the costs of childbirth are covered by a fund restricted to EU citizens and therefore, again, referrals to NGOs to finance deliveries are reported for non-EU migrants.

“Every woman who comes here for antenatal care has a gynaecologist who is responsible for her and, in parallel, a social worker. Her job is to make sure that the woman is safe. To secure the delivery, that they, if they are undocumented women, can legalise themselves through an exceptional leave to remain. That they have a place to live, that they have financial security and, of course, that the costs for childbirth are covered by the social welfare office”(P8)

As mentioned by the expert, during the legal maternity protection period, a minimum of six weeks before and eight weeks after giving birth, women can apply for a temporary suspension of expulsion (“Duldung”). Some states and cities in Germany have prolonged this period to three months before and after birth.

For pregnant women with in-patient needs, i.e., high-risk pregnancies, a major challenge in care is described: While the doctors in LPHAs try to provide as much ambulant antenatal care as possible, at some point they must send these patients to intrahospital treatments before complications arise. Thus, these cases are regularly not covered as they are not legally recognised as an emergency, which would be the only way to finance healthcare by the authorities, and therefore these women are confronted with high hospital bills and choose to self-discharge earlier, increasing the risk of complications. Expert P2 from one of the LPHAs with gynaecological services explains the challenges as follows: 

“[Uninsured] women had to be referred to the hospital during pregnancy, because they had been treated by our outpatient gynaecologists with whom we cooperate, who at some point were no longer able to assume medical responsibility […] This was not considered an emergency, although it would have led to one for the child and the mother, if not treated. Accordingly, women often discharged themselves from the hospital early. Nevertheless, they received a huge bill for several thousand euros. A consequence of this, one does not know whether it arose despite or because of the shortened stay, a stillbirth occurred. And sometimes the stillbirth was considered an emergency for curettage or stillbirth, sometimes not.”(P2)

#### 3.4.3. Chronic Disease and In-Patient Treatment

Outpatient care for the uninsured is presented by the experts as comparably easy to organise and relatively positive. However, they point out that this care is of a lower quality than standard care in the regular health care system.

“So the fact that the two drop-in centres are very well connected with doctors, who then also work free of charge, is certainly not bad, but of course it is still unsatisfactory when sick people are dependent on good will. We would prefer to have it regulated in an orderly manner. Not to belittle this commitment or saying that it is not necessary, I would say that it is a stopgap (“Notnagel”) that works quite well.”(P11)

Chronic diseases with long-term outpatient medication and diagnostic needs are described as challenging. In one setting, although there is a local fund for individual case assistance, this was only legible to acute diseases. Accordingly, in the absence of access to health insurance, the provision of care for chronic disease was undertaken by NGOs. In general, in-patient needs and, i.e., oncological diseases, are a challenge due to the high costs, which currently remain partly without a solution. Here, the cooperation via donation-financed projects is described if no access to the regular system can be established.

“And the other day there was an enquiry about treatment costs for an operation on a tumour of 12,000 euros. And then of course you must think, okay, that would somehow take 5% of my budget away in one swoop. And then you try to somehow find a solution with the [NGOs] by saying: okay, we also have services that are financed by donations, and on the other hand, you say we will contribute half of the costs and the civil society will pay the other half.”(P4)

#### 3.4.4. Mental Health

The LPHAs primarily deal with mental health of the UM within the framework of the socio-psychiatric services (“Sozialpsychiatrischer Dienst”), as well as within the framework of medical consultations on-site where available. This theme was generally very little reported during the interviews. Reported issues concerning mental health for UM were addiction, often in combination with homelessness, as well as foreign students and Au-pairs that presented mental health care needs and lack of insurance coverage for non-urgent care.

“Au pairs were a big problem in the pregnancy counselling centre. Au pairs with unbelievably bad health insurance that excluded everything that was not acute, but who were mostly pregnant or sometimes also needed psychotherapeutic treatment, so that this inadequate care should perhaps also be taken into account.”(P10)

#### 3.4.5. Socio-Legal Clearing

So-called “clearing” is a common term for socio-legal counselling to achieve systematic integration into insurance coverage for uninsured migrants in Germany. These “clearing houses” (“Clearingstellen”) [4,33] offer sufficient resources to achieve long-term solutions for a proportion of the clientele. Clearing houses can be found as grant-funded services at NGOs or as a supplement to existing offers by the authorities. The extent of knowledge and resources for clearing varies from the passive provision of information to the client to intensive case management on individual cases. Due to the increased number of EU citizens in grant-funded services or services on-site of LPHAs, clearing was set up at several locations, as at least theoretically, access to health insurance should be possible for this population. However, this is partly negated by the experts: all experts with a socio-legal mandate reported difficulties in bringing EU citizens into an insurance relationship due to complex legal obstacles. As this group of migrants does not fall under the category of “illegal” migrants, these UM were even excluded from financing from certain grants due to their theoretical access to healthcare in their home country. One expert from an SSO judged that clearing is difficult for volunteers and should be undertaken by paid experts to improve results.

For pregnant women, it is reported that when clients are advised to obtain legal resident status and thus right to social welfare, e.g., by denouncing human trafficking or the acknowledgement of paternity, they are repeatedly subjected to repression, contrary to inter-institutional agreements with the immigration authorities. The expert P8 from an LPHA with pregnancy counselling illustrates this from her experience:

“There is an employee who is responsible from the immigration authority, with whom there are meetings and agreements. Nevertheless, all that is discussed there is then actually different again and the women must run repeatedly to hearings, for example, and are sent away, are treated badly, are put down, the situation is always different from what was actually agreed on.”(P8)

Another challenge between authorities is illustrated by one expert from an LPHA: EU citizens with addiction problems are repatriated to their country of origin but ultimately return to Germany due to their local social web. In these cases, the interests of the health authorities consist in providing local care in the critical perspective of repatriation, while the immigration authority carries it out as a legally proven means. The expert, P8, goes on to describe another challenge, the highly complex situation of multi-pronged vulnerability among pregnant women from EU countries:

“Homelessness, that is a very big issue with [women from the EU]. They are actually worse off than women without a residence permit. I know that 12 weeks before they give birth they will be cared for and if the father of the child has been found by then and they can secure their stay here through the father of the child, then I know that their lives will continue. So then they will be able to stay here and build up their lives. Unfortunately, it’s quite different for people from EU countries. They are here with a legal residence permit and are allowed to live here, but it is very difficult for them to acquire a claim to social benefits or health insurance.”(P8)

### 3.5. Development of Local Care Settings

#### 3.5.1. Role of Public Authorities

The initiation of the exchange on local health care needs of UM, which ultimately contributes to the creation of public services, is attributed to local civil society actors. However, staff of local authorities are also stakeholders who participate in the design of new services to be created. For example, an anecdote is told about how, based on the case of a woman suffering from AIDS, the health authority and civil society actors ultimately created a mutual fund to finance in-patient needs.

“And with that [case] we then went out and contacted churches. I think this woman’s case has moved a lot in some people’s minds, I would say. There were a few others, but it also led to churches and hospital chaplains joining in and doing something. But the local authority was actually much of a driving force in the whole story.”(P3)

The regulatory role of managing grant-funded projects means professionalising reporting on designated projects and communicating the state of the care situation to policymakers. This role of the authority means that experts give recommendations for the development of the supply offers within the scope of their activities. In this context, the same experts take on the task of drafting a proposal for possible new projects. For this purpose, they draw data from on-site services and those of the local NGOs to obtain insight into the care situation. More often, but not exceptionally, experts who had personal involvement in NGOs prior to official work saw their role as lobbying for UM and fighting for their right to health care.

“Ultimately, this target group needs a political lobby, and someone has to make a strong case for this. We can promote this by advocating offers such as the [drop-in centre for uninsured migrants].”(P6)

Experts describe that when NGOs receive public funding and aim to expand the budget utilisation, i.e., from ambulatory to in-hospital treatments in a project with anonymous health care vouchers, the regular exchange with local authorities managing the grant can serve as direct access to political stakeholders that can facilitate the transition to publicly funded health care to UM.

#### 3.5.2. Outsourcing

LPHAs rely, at least partially, on local NGOs that provide funding, direct medical services, and volunteer medical specialists, as well as organisations that provide counselling for UM. If health care needs for UM were identified during consultations in LPHAs and further medical services were needed, referral to on-site services was provided, if possible; otherwise, referral to NGOs or physicians that provide unpaid services was common. As aforementioned, maternity programs are limited, and the outsourcing of the financial burden onto NGOs is a regular practice in these services. Another example is given as an NGO who was invited to open a walk-in clinic: the LPHA helped to find accommodation, and a former public health officer volunteered as a physician when established. Even in settings with relatively high annual public funding for the provision of healthcare for UM, government actors still rely on unpaid work by physicians and other volunteers. This was not the case in the setting where an anonymous health care voucher was being established, as there was no unpaid volunteer work in the publicly funded project for ambulatory care. At the time of the interview, however, the in-patient care was not included. The expert, P2, from an LPHA with extended services describes outsourcing as a regular and normalized practice:

“We are still trying to expand a network of specialists, because we only have general practitioners and a paediatrician here at the office. That means we try to cover all other specialties through cooperation with registered doctors with corresponding agreements where it then costs less or is partly done free of charge.”(P2)

## 4. Discussion

### 4.1. Summary of Results

The experts interviewed in this study describe a large variety of involvement models for public authorities in the parallel health care system. While the LPHAs are directly involved in providing medical services, SSOs and SPHAs are coordinating grant-funded projects and gatekeeping access to social benefits, including health insurance. The services on-site, as well as supporting and funding of NGO driven services, are the result of perceived health care needs and political will. We observed the fragmentation of services between public authorities, as well as between publicly funded and donor-funded services. The services itself that are provided by public authorities are local and goodwill-based efforts that remain inadequate: experts report insufficient funding, a lack of specialised services, as well as legal obstacles that worsen socio-legal access to health care. NGOs and other volunteers keep substituting for the lack of access to regular health care from public institutions.

### 4.2. Discussion of Results

Experts from public services report similar themes to the challenges in the health care provision for UM as NGO actors: chronic disease [34], mental health, addiction and homelessness [35], as well as in-patient needs [36]. It is common practice in LPHAs to refer cases with unresolved financial or legal challenges to NGOs, as well as to physicians who provide volunteer or reduced paid services (“Outsourcing”). In-patient needs are rarely covered and might lead to costs on the side of the hospitals [36], patients or NGOs. This dilemma of compensating for shortcomings by public actors is recognised by NGOs but difficult to resolve due to the ethically motivated intention to help patients [37]. Outsourcing to NGOs in providing health care to relieve public structures is further institutionalised by the partial funding of NGO services. The interaction of actors in the parallel health care system, resulting from reporting on funding utilisation, reporting to political actors on challenges, and the coordination of projects, can form an important local information platform. A common-interest complex between public authorities and NGOs can therefore lead to substantial lobbying for health care needs and services for UM on a government level. When drafting proposals for decision-makers, experts from public authorities profit from their experience as an intermediary, navigating between the opposing poles of the interest of traditionally restrictive immigration policies and policies on health equity. This could theoretically lead to an improvement of the local care situation, but even in settings where LPHA services were extensive and local funds and funded collaborations with NGOs existed, important gaps in coverage of the health care services for UM were described. The depth of coverage by services is low due to general resource scarcity and lack of specialised staff, and are therefore expensive services. These services are only available on a case-by-case basis as the height of coverage is low, and cost-sharing is often not an option for UM.

For example, most LPHAs in Germany provide some services in the framework of IfSG [38], but many still report difficulties in treating UM (i.e., undocumented migrants). As experts reported that even LPHAs with relatively extensive services for UM have problems in providing treatment in the context of the IfSG, this complements the reports from a survey of LPHAs in Germany from Mylius et al. in 2011 [16]: Half of the LPHAs reported that contact with undocumented migrants (24.6% of the 139 respondents) yielded some kind of treatment. Of the total of respondents, 91.6% provide counselling and testing for HIV, but according to remarks from the comments section of the survey, HIV treatment cannot be provided due to high costs. Further, a study by the German Robert Koch-Institute on HIV testing shows that “having no health insurance or medical treatment voucher decreased the odds of contact with the healthcare system more than other socio-demographic characteristics” and led to “lower odds of ever having done an HIV test than participants with health insurance”, making a case for a “health care first” approach [39]. Additionally, the public health aspect, a low threshold to access testing, and vaccination for communicable diseases (i.e., HIV, hepatitis B, and tuberculosis) are potentially more cost-effective [40,41,42].

The example of uninsured pregnant women in our study illustrates how gaps in care have traditionally been approached by adding more service fragments to an already highly-fragmented parallel health care system. Repeatedly confronted with pregnant UM, NGOs and LPHAs have created antenatal care and maternity programs that have important limitations due to limited cost coverage, especially for in-patient needs. Funds that exist are prone to restricted access due to discrimination according to residence status and country of origin. Undocumented migrant women could access health services but risk expulsion due to the necessity to expose them to immigration authorities [2]. A comprehensive literature review by Munro et al. [43] describes the challenges of undocumented pregnant women in western countries: Undocumented women accessed antenatal care less often [44,45,46] and later [44,45,47,48,49], with fewer control visits and fewer auxiliary testing [44]. Two studies [46,50] report higher risks of complications, including premature birth, lower birth weight, perinatal mortality, and admission to neonatal intensive care. In contrast, three studies have found equal birth outcomes. Although the studies included are very heterogeneous in design and population studied, we observe a potentially important public health risk for this vulnerable population.

EU migrants, especially, are searching for medical consultation within local authorities [12,13], as gaining access to insurance coverage or funding for treatment proves to be difficult in the light of the complex legal framework. From the perspective of one expert in our study, this leads to the phenomenon that it is more difficult to find funding for health care for EU citizens than for third-country residents. One report from a clearing service in Berlin indicates that not only origin but also living situation (regular address vs. homelessness) plays a major role in successful clearing outcomes: EU citizens with regular status and a permanent home address were successfully referred in 72% of males and 59% of females. The rate dropped in homeless EU citizens to 19% for males and 26% for females [14]. One study on the socio-legal clearing for the funding of in-patient treatment of uninsured patients by hospital staff demonstrates that it was more likely to find funding for third-country residents than for EU citizens, but still more likely than for undocumented patients (83.1% vs. 35.2% vs. 26.1%) [51]. It is an especially frustrating situation for the migrants themselves, as “confusing and unclear regulations play a major role in mobile EU citizens’ actual limitations of access to social benefits” [52], and Eastern European migrants seem to be generally at higher risks for negative health outcomes [53]. We, therefore, make the case that public authorities must be more involved in facilitating the access to healthcare for UM, as they can provide the professional and resource-intensive framework for the socio-legal clearing for EU citizens. The role of SSOs is therefore crucial, as the complex legal constellation needs experts to navigate challenging EU and national legislations. It was recommended by experts that the legal access for EU citizens to health insurance be simplified. Unfortunately, data protection is not applied when SSOs obtain data from UM, and the immigration authorities are informed of the assumed legal residency status, risking expulsion. As this makes it impossible for vulnerable populations to contact SSOs, especially undocumented migrants, it is necessary that effective case management is available to all UM otherwise, funded and supported by public authorities, unless legal changes are made.

If adequate public health was the goal for policymakers, as signed and ratified by Germany on the International Covenant on Economic, Social and Cultural Rights [54,55,56], UM must be able to access good quality health care (i.e., the regular health care system) in an efficient and universal manner. State-level projects such as the anonymous health voucher (“Anonymer Krankenschein”) [57] could be a means to overcome the structural fragmentation found in this study and provide widespread access to health care regardless of specific services in the scope of current national legal practice. Similar considerations have led to the introduction of the electronic health card (EHC) for asylum seekers [58], as in Germany, this population was only able to access limited health services on an ad hoc basis via treatment vouchers [6]. Despite traditional restrictions of access to the health system, scientific studies have provided evidence of a benefit in terms of health and health care access indicators [59] without an economic disadvantage for the public authorities [60]. Although evidence is in support of the nationwide implementation of an EHC for asylum seekers in Germany [59,61], this policy has not yet been adopted across all federal states due to political reasons [62]. An in-depth study on the effects of existing projects such as the anonymous health voucher could help to improve our understanding of the potential effects of more widespread implementation of such tools. At the same time, more attention should be paid to social and political reasons that hinder the large-scale implementation of policies that facilitate access, as the experience of the EHC shows that even in the presence of evidence for the benefits of improved health care access [62], concerns from the perspective of migration policy may outweigh public health considerations [63].

We did not aim to analyse a representative sample of LPHA, SPHA, and SSO, but the chosen method allows the first exploration of the perspective of the respective actors on UM. In this sample, we only found experts from authorities that are already aware of UM in their services and present a certain drive to advance projects involving this population. Therefore, the involvement and motivation for improving access to health care for UM by experts and the respective institutions they represent are likely overrepresented in this sample compared to other public authorities. There are many more public services in Germany that have UM visible among their clients that were not known at the moment of recruitment; however, we were able to study expert perspectives from a wide range of services with a variety of approaches regarding care for UM. We have not included interviews with UM in this study due to resource availability and the primary focus on public authority experts’ perspectives, although the perspective of this population must be taken into consideration in further evaluation of public authorities’ involvement.

## 5. Conclusions

This explorative qualitative study sheds light on the perspective of public authorities on the health needs and health status of UM and their role within local parallel health care systems. Experts testify that despite the important involvement of public actors in interacting with NGOs, efforts remain inadequate in regards to the depth and height of health care coverage. We thus make the case that universal access to regular health care for UM is needed to overcome the fragmentation of services. We learn that even professionals from SSO identify important legal obstacles in access to health care for EU migrants without health insurance and that highly professionalised efforts are needed for successful access improvement. Facilitating access to health care by the simplification of legal frameworks remains a primary demand by experts who exercise socio-legal counselling.

## Figures and Tables

**Figure 1 ijerph-19-06613-f001:**
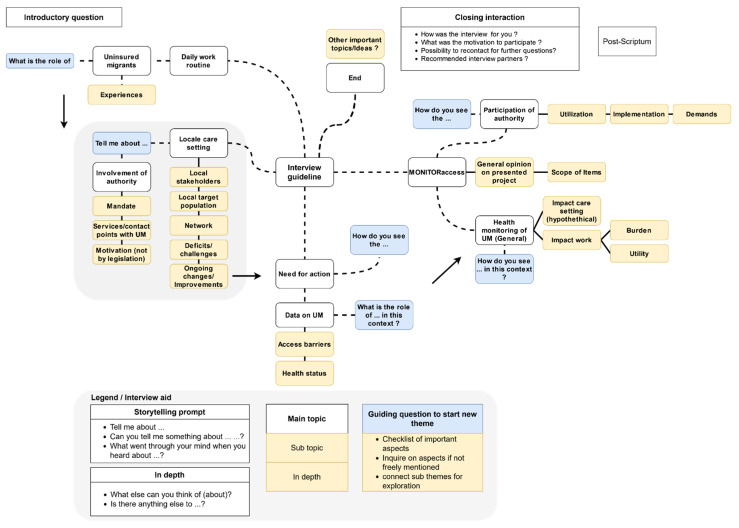
English courtesy translation of the interview guide. For the original version in German, see Appendix A. The arrows indicate the typical order of topics discussed in the interviews.

**Figure 2 ijerph-19-06613-f002:**
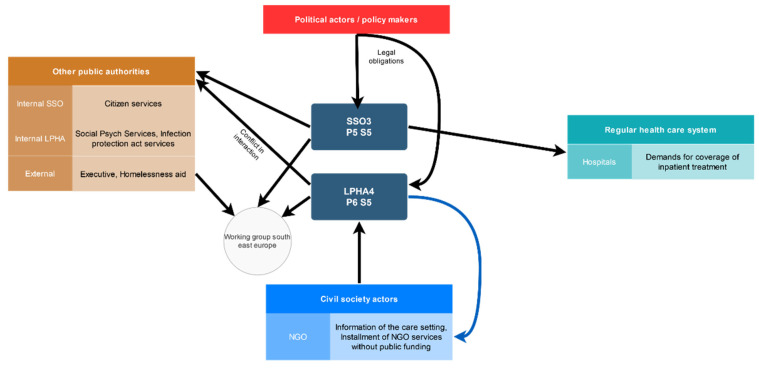
Courtesy translation of exemplary graphical presentation of the institutions and network of actors in care setting 5 (S5) as mentioned by the interviewed experts. Visualisation is to aid individual case analysis and does not present a formal systematisation of care settings. Network connections are primarily pooled from the code category “Netzwerk”. Black arrow indicates general network connection, blue arrow indicates the specific relation type “outsourcing”, based on the code “Outsourcing (Ehrenamtlich)”, indicating that a certain service is provided by an NGO or physician without complete public funding. Appendix A contains the graphical presentations of settings 1–7. Abbreviations: Local public health authority (LPHA); Social security office (SSO); interview person (P), the setting (S). Thickness of arrows does not indicate stronger/weaker relations.

**Table 1 ijerph-19-06613-t001:** Exemplary application of the interview guide from the interview with expert P6. See Figure 1 and Appendix A (German original version) for the full interview guide.

**Main topic**	How is your authority connected to the care for uninsured people here in [name of city]?
Subtopic	Now you have just named a very concrete offer where the uninsured can theoretically go, are there other contact points?
In depth	Would you say that in general there are enough offers?

**Table 2 ijerph-19-06613-t002:** Services and involvement of local public health authorities in the local health care setting with contact to uninsured migrants in the sample.

Authority	Interview Partner	Setting	Counselling—Infectious Disease	Counselling—Pregnancy	Social Psychiatric Service	Diagnostics—Infectious Disease	Diagnostics—Sexual Health/Gynaecology	Treatment—Infectious Disease	Treatment—General Medical	Birth Programme	Socio-Legal Clearing	No Data Work	Routine Data Work	Targeted Data Work	Publication of Data Work	Funding of NGO Services	Other
LPHA1	P1	S1	X	X ^1^		X		X						X			X ^2^
LPHA2	P2	S3	X	X	X	X	X	X	X ^3^	X			X	X	X		X ^4^
LPHA3	P3	S4	X	X	X	X	X	X	X ^5^		X		X	X	X		
LPHA4	P6	S5	X		X	X		X				X ^6^					
LPHA5	P8	S6	X	X		X	X	X	X ^7^	X	X ^7^		X				
LPHA6	P9	S7	X		X	X		X					X				X ^8^
LPHA7	P10	S2	X	X	X	X	X	X						X	X	X	X ^9^

Only aspects are reported that the experts mentioned in immediate reference to services for UM, therefore, underrepresentation of services is likely. The categories of services are derived from the corresponding codes (see Appendix A, page 2). Abbreviations: Local public health authority (LPHA); Interview partner (P); Setting (S); non-governmental organisation (NGO); uninsured migrants (UM). ^1^ antenatal care by volunteer doctors; ^2^ expert is participant in local NGO; ^3^ medical drop-in centre for UM on-site; ^4^ individual case decision to fund patient in maternity program; ^5^ through sexual health/gynaecology consultation on-site and outreach medical service; ^6^ no data work reported during interview, no known publication concerning UM by the LPHA; ^7^ through sexual health/gynaecology consultation; ^8^ expert volunteers officially at local NGO drop-in centre; ^9^ funding for treatment costs on individual case basis.

**Table 3 ijerph-19-06613-t003:** Summary of reported challenges in provision of health care in the work with UM by experts.

Category	Reported Challenges
In-patient needs	Expensive and lengthy treatments: oncological diseases, high-risk pregnancies, surgery, budget decisions with ethical implications
Chronic disease	Costs for regular diagnostics, cost of regular medication, i.e., HIV, hepatitis and diabetes
EU Citizens	EU foreigners in old age who live with their children who work in Germany; Multi-vulnerability: Pregnancy and homelessness, addiction and homelessness, complex legal constellation preventing access to social welfare, repeated migration
Irregular migrants	UM who do not want to expose their irregular status when financing of treatment is needed cannot be helped by public authorities
Inter-authority	Conflicts of interest between health and immigration authorities, refusal of jurisdiction between authorities
Other	Cost coverage for treatment of diseases identified in IfSG services

Abbreviations: Human immunodeficiency Virus (HIV); Uninsured migrants (UM); Infection Protection Act (IfSG).

## Data Availability

The data presented in this study are available on request from the corresponding author. The complete data are not publicly available due to privacy reasons.

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
