# Peer review of "Health Status and Access to Healthcare for Uninsured Migrants in Germany: A Qualitative Study on the Involvement of Public Authorities in Nine Cities"

_ijerph, 2022, doi:10.3390/ijerph19116613_

Round 1

Reviewer 1 Report

This is an interesting article, well structured, and presenting results and insights relevant for policy development and research.

I suggest some clarification in two aspects:

1) One issue that could be improved is on the relevance of the intersectoral nature of the topic (migration/immigration policy and regulations and health/public health policy regulations and structures) and the influence of this intersectoral nature for health care provision in a regular and / or parallel system.

2) It reads a bit confusing for the reader when the text switches between "uninsured migrants", "undocumented migrants" and/or "EU migrants (to be differentiated from third country nationals?" and it takes a lot of background knowledge to be able to understand the difference and the related consequences to health care provision. This could be either harmonised in wording or briefly explained. 

Reviewer 2 Report

This is a very interesting paper on healthcare access issues for uninsured migrants in Germany. The qualitative nature of the study , although subjective, highlights some important issues.

I only have 3 comments:

1) to give even a rough indication of the distribution of uninsured migrants by origin in Germany or in each of the cities included in the study. Clearly, the approach to solve the problems may be somehow different. Also, could you clarify whether under the mobility of EU migrants there is a provision of insurance coverage and under what conditions? 

2) it would be useful to clarify whether the proposed way forward would be to take action against the fragmentation of services at the level of each local government or by national legislation

3) please explain how the incurring hospitalisation costs for uninsured migrants are covered for the different categories of migrants. Also in the case of granted permit to stay e.g.  in the case of pregnancy, what is the duration of the stay and under what conditions?

Reviewer 3 Report

The present manuscript presents a very relevant and understudied topic. It is well written, and clear. Though, I would like the authors to consider adding some small suggestions:

Introduction:

  1. It is very understandable and complete. Though I would expect some lines about Universal Health Coverage (UHC), and a comparison of Germany vs. other European countries. Please, add some information, if considered relevant.

Methods:

  1. I would like to know why the authors did not interview UM? I think it should enrich the results and the discussion. Was it due to ethical concerns? Lack of resources? Did not consider it? It could be stated as a limitation of the study in the proper section.
  2. Also, other limitations of the present study might be declared.
  3. It would be relevant to include % of countries of origin of UM in Germany, currently and/or when the data was collected.

Results:

  1. Sometimes I feel that results and discussion are mixed in the “Results section”. I would encourage authors to review the results section and leave the discussion for the following section.
  2. As authors explore access to antenatal care services by pregnant women, I would like to know if, within their data, they have some relevant information about access to safe abortions by UM.
  3. In general, this section could be shortened.

Discussion:

  1. It could be discussed that, while results show the lack of funding/commitment to diagnose and treat infectious diseases in this/other vulnerable populations, cost-effectiveness analysis shows the advantages of providing care to all the population (apart from an ethical perspective in public health). Some references could be added in this regard, to add a strong point about UHC including UM and all populations.
  2. In general, this section could be shortened.

Conclusions:

  1. Though conclusions are very clear and important, as stated, I would encourage authors to do a more directed call to action for the challenges encountered, for this and other populations with restricted access to care in Germany, and all Europe.

Figures 1 and 2, and Supp. File 3, need to be revised for clarity (simplification of the content and format), and be submitted with higher quality.

I hope these suggestions could help authors to improve the manuscript.

Best,

Round 2

Reviewer 3 Report

Thank you for addressing all the comments.